# Enhancing Care Through a Virtual Canadian Community of Practice for Managing Immune-Related Adverse Events

**DOI:** 10.3390/curroncol32030140

**Published:** 2025-02-27

**Authors:** Khashayar Esfahani, John Walker, Kevin Bambury, Eoin O’Carroll, Stephanie Snow

**Affiliations:** 1Department of Oncology, Jewish General Hospital, McGill University, Montreal, QC H3T 1E2, Canada; 2Department of Oncology, St. Mary’s Hospital, McGill University, Montreal, QC H3T 1M5, Canada; 3Cross Cancer Institute, University of Alberta, Edmonton, AB T6G 1Z2, Canada; john.walker2@albertahealthservices.ca; 4ONCOassist, V93 T8K7 Killarney, Ireland; kbambury@portablemedicaltechnology.com (K.B.); eoin@portablemedicaltechnology.com (E.O.); 5QEII Health Sciences Centre, Dalhousie University, Halifax, NS B3H 1V8, Canada; stephanie.snow@nshealth.ca

**Keywords:** irAE, immune checkpoint inhibitor, community of practice, mobile application

## Abstract

The advent of immune checkpoint inhibitors (ICIs) has significantly transformed cancer treatment outcomes. However, these therapies can induce immune-related adverse events (irAEs) that may affect any organ system, sometimes requiring specialized expertise. As ICIs are increasingly used across various tumor types and in earlier treatment settings, not all practitioners have the necessary support network to handle complex irAEs. To address this gap, we collaborated with ONCOassist, a leading app for oncology professionals, to establish the first virtual Canadian Community of Practice (CoP) focused on irAEs. The CoP facilitates continuous learning and improves patient care among Canadian clinicians treating patients with immunotherapy by providing a platform for knowledge exchange and peer-to-peer support. This article outlines the development and growth of the CoP on irAEs, highlighting both successes and challenges. As of May 2024, over a year since its inception, the CoP on irAEs has attracted almost 130 Canadian oncology healthcare professionals, and peer-to-peer interactions and engagement continue to increase. To ensure its long-term sustainability, we plan to evolve and adapt the CoP to meet the needs of the oncology community and address clinical challenges associated with new therapies.

## 1. Introduction

The advent of immune checkpoint inhibitors (ICIs) has revolutionized cancer treatment, offering new hope and significantly improved outcomes for patients. However, these therapies can also lead to immune-related adverse events (irAEs), which require prompt recognition and effective management. Various oncology societies have issued guidelines for managing irAEs, and continuing education is available through clinical rounds, invited speakers, and even a dedicated podcast series (i-MPACT Podcast) [1,2,3,4]. Despite our growing understanding of irAE management, dysimmune toxicities can be unpredictable, and severe irAEs remain particularly challenging. In these instances, access to a network of subspeciality experts may be needed to assist oncologists in the diagnosis, treatment, and monitoring of dysimmune toxicities [5].

With ICIs being increasingly approved in multiple indications across most tumor types, irAE management has become a significant aspect of clinical practice. An audit of clinical practice from 2018 to 2020 at the Cross Cancer Institute in Edmonton demonstrated that 25% of outpatient department time was spent managing immune-related toxicities. Despite the widespread use of ICIs, toxicity management continues to be a highlighted need in educational assessments, thus forming the basis for initiating a community of practice on irAE management.

A community of practice (CoP) is a group of people who share a concern or a passion for something they do and learn how to do it better through regular interaction [6]. An effective CoP encourages open communication, collaboration, and information sharing among its members. In healthcare, CoPs can facilitate learning and the exchange of information or knowledge with the goal of improving practice [7]. While many CoPs rely on face-to-face meetings, online forums or social media platforms offer the advantage of crossing institutional and geographical boundaries.

In collaboration with ONCOassist, a point-of-care application providing clinical tools used daily by oncology healthcare professionals, we initiated a pan-Canadian virtual Community of Practice (CoP) focused on managing chronic, rare, or acute irAEs. In a resource-constrained healthcare environment, our goal was to create a platform that enables a multidisciplinary approach to irAE management. This platform allows academic and community clinicians, as well as subspecialists involved in irAE management, to exchange expertise across geographical boundaries, ultimately aiming to improve clinical management and patient outcomes.

## 2. Materials and Methods

### 2.1. Design and Development

The Canadian Community of Practice on irAEs was conceived by a steering group of clinicians (KE, JW, SS) and developed by ONCOassist. Project management support is provided by Agence Unik (Blainville, QC, Canada), with oversight by Master Clinician Alliance (MCA), a not-for-profit physician organization. ONCOassist is a free smartphone app used worldwide to aid healthcare professionals working in oncology. Key features of ONCOassist are the clinical tools oncology professionals need at the point of care (shown in Appendix A). ONCOassist was initially developed at University College Cork through the Masters in E-Business program in 2012. Since it was originally launched, ONCOassist has received wide-scale acceptance amongst oncology clinicians globally. It was promoted by the European Society of Medical Oncology and European Oncology Nursing Society and is used by more than 87K users in more than 180 countries. In Canada, at the time that this collaboration was initiated, there were about 200 Canadian ONCOassist app users. As of March 2024, there are 565 validated HCPs. The app is constantly improving based on user feedback, ensuring the latest tools and content are available to oncology HCPs. The development process for the CoP began in February 2022 and lasted 7 months. The CoP was developed in native iOS and Android, ensuring optimal usability. A beta testing phase began in Oct 2022. A group of eight medical oncologists were invited to test the CoP and provide feedback. While ONCOassist can be accessed by both the web and mobile application, the CoP on irAEs was developed for the mobile app only.

### 2.2. Launch

The CoP on irAEs went live in November 2022 to a select group of HCP beta testers and was officially launched in January 2023. To access the CoP, HCPs must download the ONCOassist application, which is freely available on the Apple App Store or Google Play for Android devices. On the home screen of ONCOassist, a “CoP on irAEs” icon is featured as a new tool. Access to the CoP is gated, and first-time users must provide their medical license number and province of practice. HCP credentials are manually validated by ONCOassist within 24 hrs. Once validated, HCPs have access to exclusive content and are held by the CoP code of conduct. Patient identifiers must not be used; urgent irAE treatment management advice should not rely only on the CoP, and HCPs are responsible for reporting any adverse events to regulatory authorities, as they do in practice. Employees of a pharmaceutical/biotech company or a for-profit organization working in the healthcare environment (such as a communication or marketing agency) may not request membership, even if they hold a medical license. Initially, a short survey was requested prior to gaining access to the CoP. This was later withdrawn to facilitate access. Initially, efforts to raise awareness about the Community of Practice (CoP) focused on physicians. Subsequently, the scope of awareness campaigns expanded to encompass all healthcare professionals within the multidisciplinary team, particularly pharmacists and nurses. This project was evaluated by Alberta Innovates using the ARECCI (A pRoject Ethics Community Consensus Initiative) tool and was found to be of minimal risk (see Appendix A). The project did not require ethical board approval.

## 3. Results

The CoP on irAEs was launched in January 2023. Upon first accessing the CoP, healthcare professionals (HCPs) are invited to complete an optional survey to self-assess their comfort level in managing irAEs on a scale from zero to ten, with ten indicating the highest level of comfort. A total of 115 HCPs responded to this survey question. The majority rated their comfort level around the midpoint of the scale (Figure 1A). Notably, 26% of respondents rated their comfort level below 5, while only 4% rated their comfort level as 9 or 10. Respondents identified neurologic, cardiac, and ocular irAEs as the most challenging to manage (Figure 1B). Interestingly, irAEs most frequently associated with IO therapies, such as skin and gastrointestinal issues, were rated as the least challenging to manage by the respondents.

For the initial 8 months after launch, new CoP members averaged five per month. With sustained awareness efforts, monthly membership grew and currently averages about 10 new members monthly. As of May 2024, there were 129 validated HCP members (Figure 2), with increasing activity and engagement on the CoP, as measured by the number of CoP question clicks by unique users. In 2023, a total of 59 unique users generated 614 clicks on questions or comments. Engagement will be surpassed this year since data collected from January to May 2024 reveal that already 61 unique users generated 565 clicks on questions or comments.

The CoP includes members from nearly all regions of Canada, with Quebec, Ontario, British Columbia, and Alberta accounting for over 85% of the membership (Figure 3A). Although the CoP is open to any licensed healthcare professional, initial awareness efforts were primarily directed towards medical oncologists. In October 2023, we expanded our outreach to ensure that key members of the multidisciplinary team, including general practitioners in oncology (GPOs), nurses, and pharmacists, had the opportunity to join the discussions. Currently, 69% of CoP members are medical oncologists (Figure 3B). The remaining members are predominantly nurses, pharmacists, and trainees. Upon downloading ONCOassist, users are asked to indicate all oncology therapeutic areas in which they practice. As shown in Figure 3C, the cancer types most commonly treated by CoP members are melanoma, genitourinary (GU), gastrointestinal (GI), and lung cancer. This distribution aligns well with the cancer types for which immune–oncology (IO) regimens, particularly dual IO therapies, are approved by Health Canada.

The CoP can be accessed by downloading the ONCOassist app and selecting the CoP on irAEs. The QR code in Figure 4A can be used to download ONCOassist on either Android or Apple mobile devices. The interface resembles social media platforms, allowing members to post questions or comments, reply to, and like other posts (Figure 4B). When posting a comment, members can also select a label for the irAE category, enabling them to sort, select, or receive notifications for specific irAEs. On average, responses to posted questions are provided within 24–72 h. Subspecialists are also invited to join the CoP and share their perspectives during focused discussion sessions, which last several weeks, and feature contributions from experts such as dermatologists and rheumatologists.

## 4. Discussion

Since the surge in investment in digital startups in 2018, over 325,000 mobile health applications have been developed [8]. A recent analysis of the mobile health app landscape in oncology identified 257 oncology-specific, English-language apps that meet basic quality standards. Of these, 45% were geared towards healthcare professionals, with education being the primary intended function [8]. Given the growing volume of information required for cancer treatment, mobile apps can significantly enhance the efficiency of busy clinicians [9]. ONCOassist is an oncology mobile health app designed as a point-of-care tool for healthcare professionals. Its key features include staging tools, calculators, toxicity criteria, treatment protocols, drug dosing, and medication interactions. ONCOassist was selected as the platform for building a CoP on irAEs due to its widespread use among Canadian HCPs, frequent updates, and the involvement of oncology stakeholders in its development.

In the expansive field of oncology, professionals often form structured or grassroots networks focused on a common practice area (e.g., disease site groups) or area of expertise (e.g., pediatric oncology), which function similarly to CoPs. The American Society of Clinical Oncology (ASCO) currently supports three international CoPs dedicated to clinician medical educators, geriatric oncology specialists, and palliative care clinicians [10]. Members of these CoPs can connect virtually through the ASCO myConnection Platform, where they can initiate and respond to online discussion posts, download educational resources, and build an online community [11]. Within Cancer Care Ontario, multidisciplinary CoPs exist to encourage innovation and knowledge transfer and provide a sense of community among practitioners [12].

The CoP on irAEs was created to address the need for peer-to-peer and organ specialist support in managing rare or challenging irAEs. Despite the widespread use of I-O therapies and the availability of toxicity management guidelines, a survey of 115 HCPs conducted prior to joining the CoP on irAEs revealed that there are still opportunities to improve comfort levels in managing I-O-induced toxicities. The ubiquitous use of mobile technology, particularly smartphones in clinical settings, provides an ideal platform for developing a virtual CoP. To our knowledge, this is the first HCP-only virtual network dedicated to supporting irAE management, available on an oncology-specific mobile app.

Membership in the Canadian CoP on irAEs has been steadily growing over the past year thanks to active awareness initiatives. Word-of-mouth promotion by the steering committee, particularly during speaking engagements, has played a significant role in generating interest. Additionally, a short video was presented at the Canadian Association for Medical Oncologists (CAMO) 2023 annual conference, and postcards were developed by the project management team of Agence Unik in collaboration with MCA to reach Canadian oncology professionals. HCPs were provided with information on how to download the ONCOassist app on their smartphones. ONCOassist contributed to raising awareness through targeted emails to Canadian members and LinkedIn efforts. A weekly email with the latest discussions is also sent to all validated users by ONCOassist.

Access to the CoP is gated, requiring a 24 h turnaround time for medical credential validation prior to granting access. Once credentials are validated by ONCOassist, an email is sent to the HCP to inform them of their access. First-time CoP users are asked to complete an optional three-question survey. However, the delay in access and the initial survey posed a barrier for some clinicians. Since the gated verification step could not be removed, we opted to eliminate the survey to streamline access.

Although research shows that clinicians generally have a positive attitude toward mobile health technologies, barriers to adoption persist. Perceived ease of use and perceived usefulness are critical factors influencing technology adoption [13]. We have aimed to address these factors through the user-friendly design of the ONCOassist app and the simplified CoP feature. For example, once HCP credentials are validated, access to the CoP is streamlined by not requiring a password or additional login. However, one additional barrier remains: the CoP is currently only available on mobile devices and not on the web version of ONCOassist. Despite the ease of use, skepticism towards innovative technologies among HCPs impacts their adoption and consistent use.

Sustaining a virtual CoP presents several challenges that must be addressed to ensure long-term success and effectiveness. One primary challenge is maintaining high levels of engagement and active participation from members. In virtual environments, individuals may feel less obligated to contribute regularly or may become passive observers rather than active collaborators. Strategies such as regular prompts for discussions, incentives for participation, and clear expectations for member involvement are necessary. Themed discussions with invited subspecialists (ex., dermatologists, rheumatologists) have been helpful in stimulating discussions. Despite these initiatives, widespread participation remains a challenge, similar to other social platforms. The 90-9-1 principle, where 90% of community members observe without participating, 9% contribute sparingly, and 1% (super-users) create the majority of new content, is reflected in this virtual CoP [14]. Content and comments by super-users, primarily from the steering committee, are essential to sustaining the online community [15]. We expect that additional super-users will emerge as the community grows.

Building trust and strong interpersonal relationships are essential for effective knowledge-sharing and collaboration. In virtual settings, where face-to-face interactions are limited, establishing and maintaining trust can be more challenging. The CoP on irAEs spans a large geographic region, which can result in differences in language, communication styles, and cultural norms. While all posts and comments to date have been in English, members are encouraged to post in either English or French. Steering Committee members or invited guests are available to respond in both of Canada’s official languages, thereby ensuring equitable participation and meaningful engagement.

Sustainability and evolution are additional key considerations. While the current focus of this CoP is on irAEs, we recognize that the CoP could evolve to address other clinical needs and become a vibrant hub of knowledge sharing and professional collaboration beyond I-O toxicity management. We are encouraged by the growing participation and sustained interest of our medical colleagues in this CoP on irAEs. As a future initiative, we aim to qualitatively evaluate the impact on patient outcomes through follow-up surveys sent to the CoP members through the app interface. The survey will include questions to assess whether HCP comfort levels in managing irAEs have improved due to their participation in the CoP and will also collect HCP anecdotes on patient impact where possible. This approach will provide insights into how the CoP has influenced practice. Additionally, the survey will be used to gather feedback on how the CoP can be enhanced for sustained engagement and explore whether it should evolve to address other care gaps observed in clinical practice.

## 5. Conclusions

We have demonstrated the feasibility of a pan-Canadian virtual CoP as a valuable feature within an oncology mobile app, enabling HCPs to exchange expertise across geographical boundaries. The CoP on irAEs supports collaborative learning, enhances knowledge dissemination, and promotes a multidisciplinary approach to irAE management. In resource-constrained healthcare systems, online communities may offer a potentially accessible, wide-reaching, and cost-effective intervention to facilitate clinical management and improve patient outcomes. As this CoP grows and matures, future efforts will focus on understanding the clinical impact and addressing evolving needs to sustain engagement.

## Figures and Tables

**Figure 1 curroncol-32-00140-f001:**
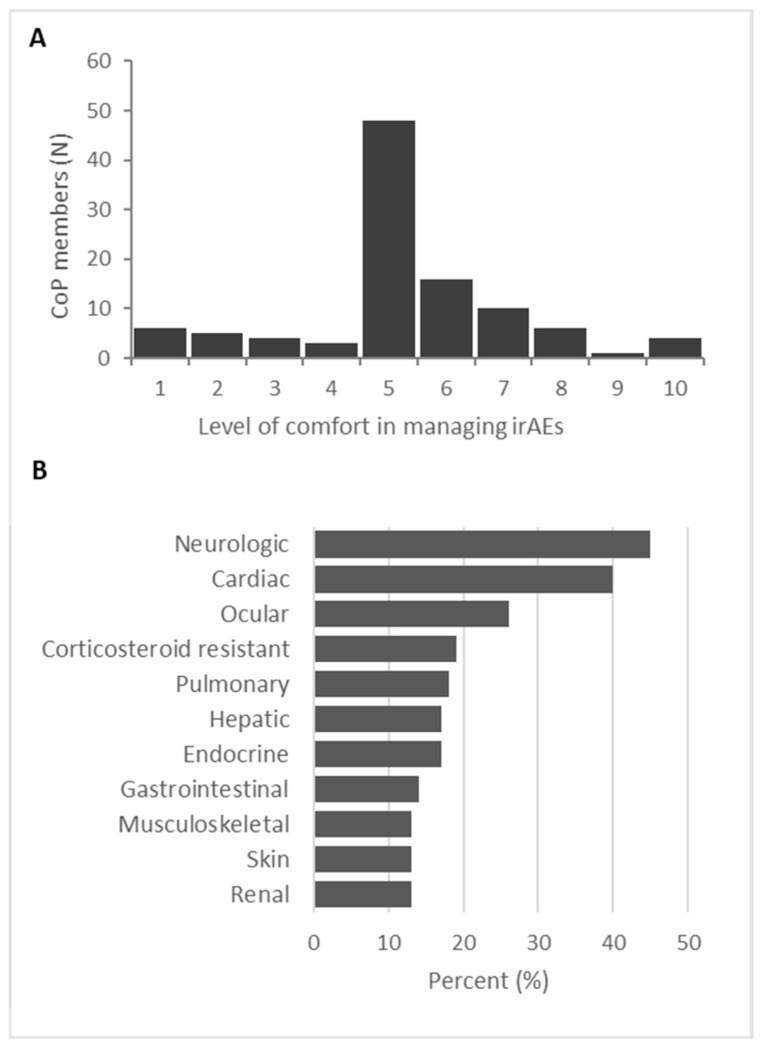
Survey questions answered by HCPs before accessing the CoP (n = 115). New CoP users were asked to (**A**) rate their comfort in managing irAEs, with zero being the lowest and ten being the most comfortable, and (**B**) identify which irAE clinical context they find challenging to manage.

**Figure 2 curroncol-32-00140-f002:**
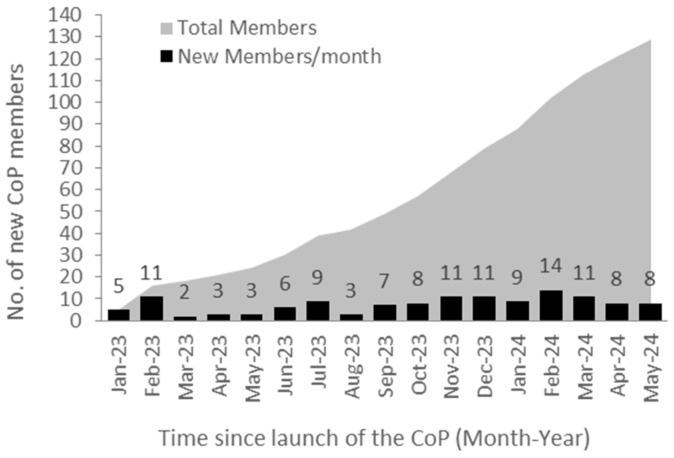
Monthly adoption rate of the COP. The black bars represent new members joining the COP monthly; the grey shaded area represents the total number of actual members to date.

**Figure 3 curroncol-32-00140-f003:**
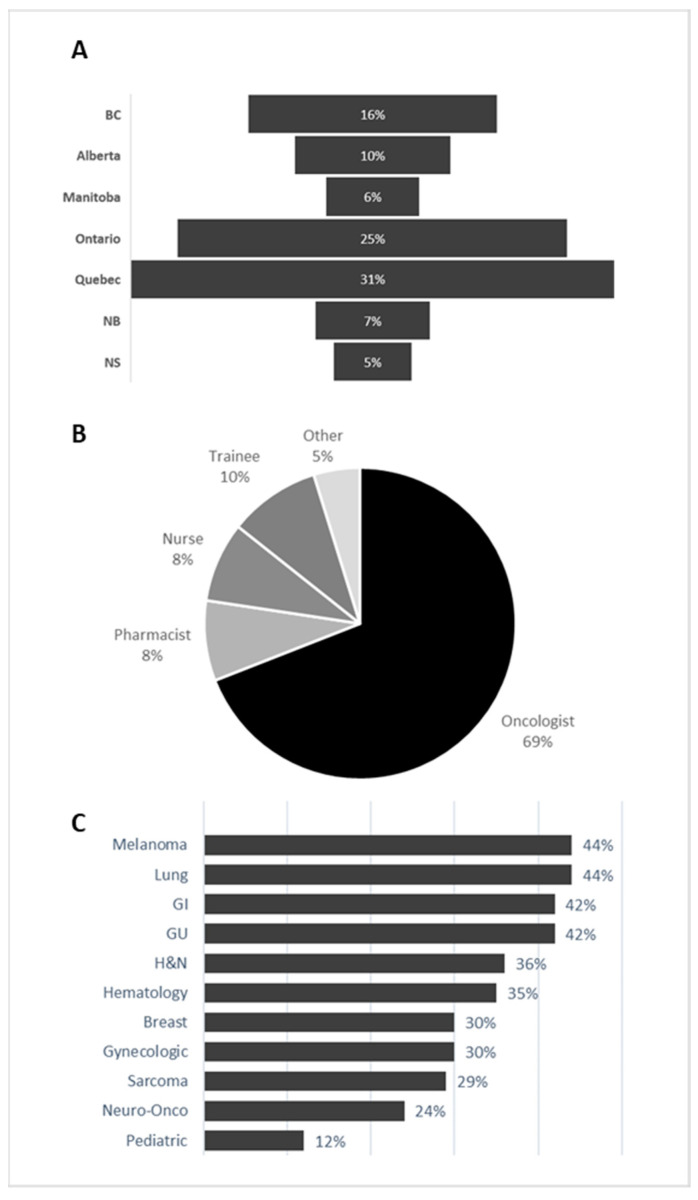
CoP members (**A**) by province, (**B**) profession, and (**C**) tumor type treated.

**Figure 4 curroncol-32-00140-f004:**
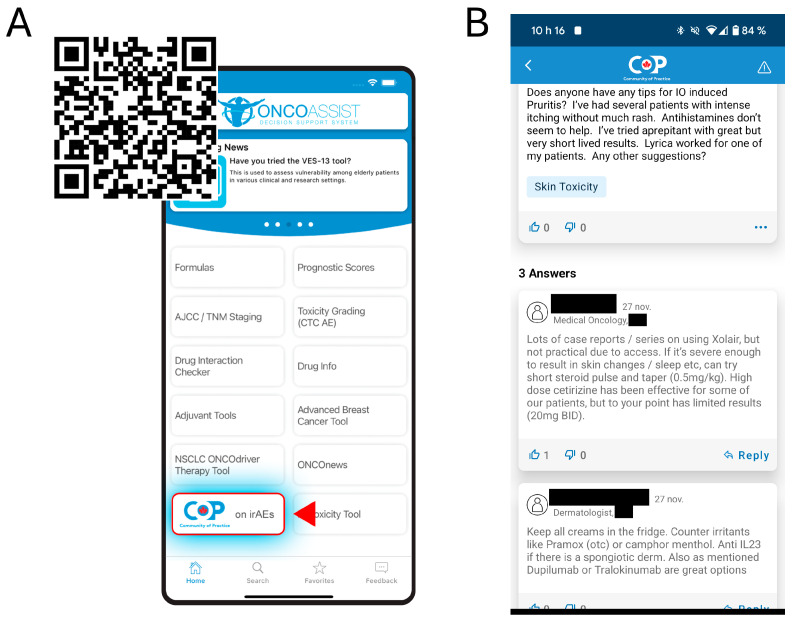
(**A**) QR code to download ONCOassist. To access the CoP, users must click on the icon “CoP on irAEs”. First-time users must provide their medical license number and province of practice. HCP credentials are manually validated by ONCOassist within 24 h. (**B**) Snapshot of the CoP interface and peer-to-peer interaction.

## Data Availability

The original contributions presented in this study are included in the article/Appendix A. Further inquiries can be directed to the corresponding author.

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
