# Peer review of "Enhancing Care Through a Virtual Canadian Community of Practice for Managing Immune-Related Adverse Events"

_curroncol, 2025, doi:10.3390/curroncol32030140_

Round 1
Reviewer 1 Report
Comments and Suggestions for Authors
Despite our growing understanding of immune related adverse event (irAE) management, severe irAEs remain particularly challenging. Since not all practitioners have the necessary support network to handle complex irAEs, the authors established the first virtual network, the Canadian Community of Practice (CoP) on irAEs to provide a platform for knowledge exchange and peer-to-peer support. To this end, the authors collaborated with ONCOassist, which is a leading oncology-specific mobile application providing clinical tools used daily by oncology healthcare professionals. The CoP on irAEs has already attracted close to 130 Canadian oncology healthcare professionals. This article outlines the growth of the CoP on irAEs highlighting both successes and challenges. In resource-constrained healthcare systems, online communities offer a cost-effective intervention to facilitate clinical management and improve patient outcomes.
This paper is relevant and worthy for publication. In order to improve patient outcomes, the reviewer suggests including the paper of Kleef, R., et al. (Cancer Immunol Immunother 70, 1393–1403 (2021). https://doi.org/10.1007/s00262-020-02751-0) who translated the GVHD theory of irAEs into an ultra-low‐dose ICI protocol containing ipilimumab (0.3 mg/kg) and nivolumab (0.5 mg/kg) in combination with other T cell stimulating modalities. The ultra-low‐dose ICI protocol proved to be safer than registered ones without compromising efficacy in 131 unselected stage IV cancer patients with 23 different cancer types who exhausted all conventional treatments. Only 8.4% of patients had grade 3 or 4 irAEs but no one died, while a 19.3 months median OS was achieved.
Author Response
Thank you for your comments.
Although we find the referenced paper interesting, the scope of this paper is not to discuss experimental trials without an approved Canadian indication which can immediately impact the incidence of irAEs.
Reviewer 2 Report
Comments and Suggestions for Authors
This manuscript presented The CoP on irAEs, an app-based physician community to share experiences of irAE management. This is an interesting and meaningful study. The background is well-introduced, and the platform is presented in detail. It can be improved by addressing the following concerns.
- Survey Follow-Up and Comparison: The authors showed survey responses indicating how 115 healthcare professionals (HCPs) rated their comfort level in managing irAEs upon first accessing this app. However, it would be valuable to know how many of these HCPs continued using the app and how they would rate their comfort level in managing irAEs after using the app for two years. Comparing these results could demonstrate how the app has helped or improved HCPs in managing irAEs over time.
- Clarification of Figure 2: The label presenting the time by month in Figure 2 should be in English for better readability and understanding by a broader audience.
Addressing these concerns would enhance the manuscript by providing additional insights into the long-term impact of the app and improving clarity in the presentation of results.
Author Response
Thank you for your insightful comments.
To point 1: This is the second phase of our project which will be completed with an update published at a later time. We have expanded the discussion to reflect this point.
To point 2: It's unclear what you are referring to. Figure 2 is already in English.
Reviewer 3 Report
Comments and Suggestions for Authors
The manuscript provides an overview of the creation and growth of the first virtual Community of Practice (CoP) in Canada focused on immune-related adverse events (irAEs) induced by immune checkpoint inhibitors (ICIs). The paper discusses the design, membership growth, and engagement of the CoP through the ONCOassist mobile application, emphasizing its role in supporting healthcare professionals in managing these complex toxicities. The manuscript is well-supported by relevant data on user engagement and the survey results from HCPs, providing valuable insights into the challenges faced by clinicians in managing irAEs. The study offers an innovative approach to fostering collaboration and knowledge sharing among oncology professionals.
There are a few areas in the manuscript that would benefit from further refinement:
- Impact on Clinical Practice: While the survey results provide useful insights into the comfort levels of clinicians managing irAEs, the manuscript could benefit from a deeper analysis of how participation in the CoP has impacted clinical decision-making or patient management. Providing specific examples or qualitative feedback from users about how the CoP has influenced their practice would add significant value to the discussion.
- Engagement and Sustainability: The manuscript mentions that the CoP has been growing steadily in terms of membership, but there is limited discussion on how the authors plan to sustain and expand the platform in the long term. More details on strategies for maintaining high engagement levels and ensuring long-term participation would be beneficial.
In summary, the manuscript provides a valuable contribution to the growing body of knowledge on virtual CoPs in healthcare, particularly in the context of managing irAEs in oncology. Addressing the areas above would further strengthen the manuscript, making it an even more compelling resource for those interested in the role of digital platforms in clinical education and support. With these revisions, the paper would be well-positioned for publication.
Author Response
Thank you for your insightful comments.
To answer both points: The impact on clinical practice is a later phase of this project, which will be completed in the future. We have expanded the discussion to reflect what this will entail. We have also expanded on our vision for sustainability